# Rheological and Physicochemical Properties of Oleogel with Esterified Rice Flour and Its Suitability as a Fat Replacer

**DOI:** 10.3390/foods11020242

**Published:** 2022-01-17

**Authors:** U-hui Kwon, Yoon Hyuk Chang

**Affiliations:** Bionanocomposite Research Center, Department of Food and Nutrition, Kyung Hee University, Seoul 02447, Korea; uhui33@khu.ac.kr

**Keywords:** oleogel, emulsion, esterified rice flour, rheological property, fat replacer

## Abstract

The objectives of this study were to produce oleogel using esterified rice flour with citric acid (ERCA), to evaluate physicochemical and rheological properties of oleogels, and to investigate their suitability as a fat replacer. Rice flour was esterified with citric acid (30%, *w*/*w*) to produce ERCA. Emulsions and oleogels were prepared with different concentrations (0, 5, 10, and 15%, *w*/*w*) of ERCA. In the steady shear rheological analysis, it was found that the values of apparent viscosity (*η*_a_, 100) and consistency index (K) of emulsions were significantly increased by increasing the concentrations of ERCA. Oleogels were prepared with different concentrations (0, 5, 10, and 15%, *w*/*w*) of ERCA. All oleogels showed a hydrophobic carbonyl bond in the Fourier transform infrared (FT-IR) spectra. The peaks on new hydrogen bonds and amorphous regions, which did not appear in oleogel prepared with 0% ERCA, were observed at 3300–3400 cm^−1^ and 1018 cm^−1^, respectively, in oleogels prepared with ERCA. With the increase in ERCA concentrations in oleogels, oil loss values were significantly decreased. In a time-dependent test, it was found that all oleogels exhibited thixotropic properties. The frequency sweep test revealed that storage modulus (G′), loss modulus (G″), and complex viscosity (*η**) values of oleogels were elevated with an increase in the concentration of ERCA. Oleogels prepared with 15% ERCA exhibited the lowest peroxide, *p*-Anisidine, and Total Oxidation(TOTOX) values. The addition of oleogels to cookies did not considerably affect appearance. However, it increased the content of unsaturated fatty acid. These results indicate that oleogels prepared with ERCA can be used as a fat replacer in food industry.

## 1. Introduction

Oleogel refers to a solid like substance with liquid oil packed into a three-dimensional network by a small amount of gelator [1]. It can be produced with various gelling agents, such as wax, wax esters, phytosterols, lecithin, polysaccharides, and proteins with various methods [2]. A promising process for the production of oleogels is the use of emulsion particles, such as an internal gelator and a surface coating material [3,4]. Emulsion methods for oleogelation can form tightly packed oil droplets and semi-solid materials to inhibit lipid oxidation [4,5].

Solid fats are crucial ingredients in the food industry since they play an important role in enhancing the hardness, viscosity, spreadability, plasticity, and flavor of high fat food products. However, because most solid fats are composed of saturated and trans fatty acids, excess intake of those fatty acids could negatively affect human health, leading to obesity, diabetes, and cardiovascular diseases. Accordingly, in recent years, studies on replacing solid fats with edible oleogels (made up of unsaturated vegetable oils) have been conducted by many researchers [6].

Rice flour or starch cannot be generally used as an emulsifier due to its high hydrophilicity [7]. Thus, it needs to be modified for providing hydrophobicity to increase emulsion stability. To enhance the emulsion stability and rheological property, rice flour can be subjected to physical, chemical, and enzymatic modifications [8]. Chemically modified flour or starch can be obtained by various chemical reactions including etherification, esterification, and dual modification [9]. Previous studies reported that rice flour or starch esterified with dicarboxylic acids, such as octenyl succinic acid, oxalic acid, and citric acid, has been used as an emulsion stabilizer since the hydroxyl groups of modified starch were replaced by the hydrophobic carbonyl groups, consequently enhancing its emulsifying stability [10,11]. Although the development of esterified corn starch, cassava starch, potato starch, and rice starch with citric acid has been reported earlier [10,12], there has been no study on esterified rice flour with citric acid (ERCA).

A few studies have investigated the formation of oleogel using polysaccharides such as guar gum, Arabic gum, fenugreek gum, citrus pectin, cellulose, and hydroxypropyl methyl cellulose as emulsifiers [3,5,13,14]. However, until now, no studies have elucidated the utilization of ERCA for producing oleogels. In this study, we hypothesized that ERCA could be used as emulsion stabilizer and coating material due to its hydrophobic carbonyl groups. Moreover, the development of oleogel using ERCA would hinder oil loss but improve the rheological properties and oxidative stability more efficiently than oleogel without ERCA. Therefore, the aims of this study were (1) to produce emulsion and oleogels prepared with different concentrations of ERCA as an emulsifier or coating material, (2) to evaluate physicochemical and rheological properties of oleogels, and (3) to investigate their potential for application in the production of cookies.

## 2. Materials and Methods

### 2.1. Materials

Rice flour and citric acid anhydrous were purchased from Nongshim Co., Ltd. (Seoul, Korea) and Junsei Chemical Co., Inc (Tokyo, Japan), respectively. Soybean oil and beeswax were purchased from CJ Co. (Seoul, Korea) and Daejung Chemicals & Materials Co. (Siheung, Korea), respectively. Cumene hydroperoxide and *p*-Anisidine were purchased from Sigma-aldrich Chemical Co. (St. Louis, MO, USA). Other chemical reagents were of analytical grade.

### 2.2. Preparation of Esterified Rice Flour with Citric Acid

ERCA was obtained based on the procedure of Klaushofer et al. (1978) [15] with some modifications. Briefly, 15 g of citric acid anhydrous was mixed with distilled water (100 mL) and pH of the mixture was adjusted to 3.5 with 10 M NaOH. Rice flour (50 g) was mixed with citric acid solution (100 mL) and equilibrated by stirring for 16 h at room temperature. The dispersion was subsequently dried in a drying oven at 60 °C to have a moisture level of 5–10% (*w*/*w*). The dried mixture was ground and cured in a drying oven at 140 °C for 4 h and rinsed 3 times with distilled water to remove unreacted citric acid. The washed sample was dried for 24 h at 45 °C and ground to pass through a 150 µm sieve. The rice flour esterified with citric acid was named ERCA.

### 2.3. Characterization of Emulsions

#### 2.3.1. Preparation of Emulsions

Emulsions were obtained by a procedure described by Luo et al. (2019) [14] with some modifications. Water phases (200 g) were prepared by dispersing ERCA with different concentrations (0, 5, 10, and 15%, *w*/*w*) in distilled water for 10 min at 70 °C. Oil phases were obtained by dissolving beeswax (1 g) in soybean oil (49 g) at 70 °C for 10 min. Oil phases were mixed with water phases. Emulsions were obtained after homogenizing the mixture of oil and water phases using a homogenizer (Velp Scientifica, Usmate, Milano, Italy) at 20,000 rpm for 2 min.

#### 2.3.2. Rheological Properties of Emulsions

##### Steady Shear Rheological Properties

Steady shear rheological properties of emulsions prepared with different concentrations of ERCA (5, 10, and 15%, *w*/*w*) were determined with a rheometer (MCR 102, Anton Paar, Graz, Austria) using a parallel plate with a diameter of 50 mm at 70 °C. The geometry gap was set at 100 μm. Steady shear data were obtained over a shear rate rage of 0.01–1000 s^−1^. To describe flow properties of each sample, shear stress-shear rate data were fitted to the power law model (1).
(1)$$\sigma =K\times {\dot{\gamma }}^{\;n}$$
where *σ* was the shear stress (Pa), $\dot{\gamma }$ was the shear rate (s^−1^), K was the consistency index (Pa·s^n^), and *n* was the flow behavior index. The apparent viscosity (*η*_a_, 100) at 100 s^−1^ was calculated using magnitudes of *n* and K.

##### Frequency Sweep

A frequency sweep test of emulsions prepared with ERCA (5, 10, and 15%) was carried out to elucidate the storage modulus (G′) and loss modulus (G″) of emulsions. Frequency sweep tests were carried out between 0.628 and 62.8 rad/s of angular frequency at a strain value of 1% (linear viscoelastic region).

### 2.4. Preparation of Oleogels

Oleogels were produced based on the procedure described by Luo et al. (2019) [14] with some modifications. To produce oleogel samples with different concentrations of ERCA (0, 5, 10, and 15%), emulsions were immediately cooled at −70 °C and dried in a freeze-dryer for 48 h to remove the water phase.

### 2.5. Characterization of Oleogels

#### 2.5.1. FT-IR Analysis

ATR-Fourier transform infrared (ATR-FTIR) spectrum was measured with a Fourier transform infrared spectrophotometer (TENSOR27, Bruker, Germany). Analysis was performed at room temperature. Spectral scanning ranges were 4000 to 500 cm^−1^.

#### 2.5.2. Oil Loss

Oil loss values of oleogel samples were evaluated based on the method described by Wijaya et al. (2019) [16] with some modifications. Briefly, oleogel (1 g) was centrifuged in a 50 mL conical tube at 3500 rpm for 15 min. Weights of the conical tubes were measured before and after the addition of oleogels so that actual sample weights were obtained. After centrifugation, released oil at the surface was removed and weighed. Oil loss values were calculated with the following Equation (2).
Oil loss value (%) = (w_1_ − w_2_)/w_1_ × 100(2)
where w_1_ was the weight of the initial sample, and w_2_ was the weight after removing the excess oil.

### 2.6. Rheological Properties of Oleogels

#### 2.6.1. Time-Dependent Properties

A time-dependent fluid measurement of each oleogel was performed using a rheometer with a parallel plate (diameter: 50 mm, gap: 400 μm) at 25 °C. Time-dependent properties of samples were carried out under alternating shear rates (0.1, 10, and 0.1 s^−1^). Samples were molded into a disk-shape using a molder (25 mm inner diameter and 2 mm height; Dongwon Co., Incheon, Korea).

#### 2.6.2. Frequency Sweep

Frequency sweep tests were performed at an angular frequency (*ω*) of 0.628 to 62.8 rad/s with strain value of 0.5% (linear viscoelastic region) at 25 °C to elucidate storage modulus (G′) and loss modulus (G″).

#### 2.6.3. Temperature Sweep

Temperature sweep tests were performed at a strain 0.5% and a fixed angular frequency at 6.28 rad/s. Oleogels were transferred to the rheometer plate at 4 °C. The exposed sample edge was covered with a thin layer of light paraffin oil to prevent evaporation during heating and cooling. Heating and cooling steps of the program contained temperature elevation from 4 °C to 96 °C and reduction from 96 °C to 4 °C, respectively, at a heating or cooling rate of 4 °C/min.

### 2.7. Oxidative Stability of Oleogels

#### 2.7.1. Accelerated Oxidation

To evaluate the oxidative stability, all samples were subjected to accelerated oxidation by thermal treatment performed in a drying oven at 140 °C for 0, 1, and 2 h.

#### 2.7.2. Peroxide Value

As a marker of primary products of lipid oxidation, peroxide value (PA) was measured with a method described by Diaz et al. (2003) [17] with some modifications. Lipid hydroperoxides were extracted by mixing 0.3 g of samples with 1.5 mL of 3:1 isooctane/2-propanol, vortexed, and centrifuged for 2 min at 1000× *g*. The upper layer (0.2 mL) was collected and 2.8 mL of 2:1 methanol/butanol was added. Then 30 µL of thiocyanate-ferrous solution was added as an indicator and then absorbance was measured at 510 nm using a spectrophotometer (Multiskan Go Spectrophotometer, Thermo Scientific, Vantaa, Finland). Hydroperoxide contents were determined using a standard curve prepared with known concentrations of cumene hydroperoxides (0–100 mM).

#### 2.7.3. *p*-Anisidine Value

The *p*-Anisidine value (AV) was measured according to AOCS method Cd 18–90 (AOCS, 2004) [18].

#### 2.7.4. Total Oxidation (TOTOX) Value

Total oxidation (TOTOX) values of each sample were calculated based on peroxide value (PV) and *p*-Anisidine value (AV) using the follow Equation (3).
TOTOX = 2PV + AV(3)

### 2.8. Characterization of Cookies with Oleogels Prepared by ERCA

#### 2.8.1. Preparation of Cookies

Cookies were prepared using the AACC method 10–52 (2000) [19] with some modifications. Different cookie formulations were applied using shortening and oleogel/shortening blends at the ratio of 50:50. First, cream was prepared by blending shortening or oleogel/shortening blends (40% of wheat flour) with sugar (45% of wheat flour), skim milk (1.5% of wheat flour), and baking powder (0.5% of wheat flour) by using a mixer at speed 2 for 2 min. After scraping down, the cream was mixed at speed 4 for 1 min and mixed with water at speed 2 for 2 min. Wheat flour was then added to the cream mixture and mixed at speed 2 for 2 min. The cookie dough was sheeted to a thickness of 10 mm and cut to a diameter of 50 mm. After baking at 180 °C for 10 min, cookies were cooled for 60 min.

#### 2.8.2. Dimensional Characteristics of Cookies

Dimensional properties such as thickness, width, and spread ratio of cookies were evaluated. After cooling of cookies for 60 min, thickness (T mm), width (W mm) measurements of the cookie samples were taken using a caliper. The spread ratio (W/T) was calculated by dividing the width by the thickness.

The cookie samples were weighed, and loss rate was calculated using the follow Equation (4).
Loss rate (%) = 1 − (w_1_/w_2_) × 100(4)
where w_1_ was the weight of cookie samples, and w_2_ was the weight of batter before cooking.

#### 2.8.3. Fatty Acids Compositions of Cookies

Fatty acid compositions of cookie samples were determined using a gas chromatography with flame ionization detector (GC-FID) system (7890A, Agilent Technologies, Santa Clara, CA, USA). Lipids of cookies were extracted with ethyl ether (200 mL) for 12 h. Each lipid sample was methylated with 2 mL of Boron trifluoride-methanol reagent and 1 mL of toluene. Lipids of cookies were injected into a SP-2560 capillary column (100 m 0.25 mm 0.2 um, Supelco, Bellefonte, PA, USA). The oven temperature was programmed to be 100 °C for 4 min and raised to 240 °C (3 °C/min) which was held for 20 min. Helium was used as carrier gas and commercial fatty acid standards (37 component FAME Mix) were obtained from Supelco (Bellefonte, PA, USA). Fatty acid compositions are expressed as the percentage of the total peak area from all methyl esters.

### 2.9. Statistical Analysis

All statistical analyses were conducted using SAS version 9.4 (SAS Institute Inc., Cary, NC, USA). Analysis of variance (ANOVA) was performed with the general linear models (GLM) procedure to elucidate significant differences among samples. Means were compared using Fisher’s least significant difference (LSD) procedure. Significance was defined at 5% level.

## 3. Results and Dicussion

### 3.1. Characterization of Emulsions

#### 3.1.1. Steady Shear Rheological Properties

Emulsions were subjected to steady shear rheological measurements to evaluate the effect of ERCA concentrations (5, 10, and 15%) on the emulsion. Because emulsion without ERCA had water-like properties and became destabilized rapidly on the plate of the rheometer, the steady shear rheological properties and frequency sweep test of the emulsion without ERCA were not performed.

As shown in Figure 1, an increase in ERCA concentration resulted in higher shear stress of emulsions when shear rates were the same. Higher values of shear stress indicated an improvement in the emulsifying stability of the emulsion [20]. Therefore, the emulsion prepared with ERCA 15% is the most stable.

The analysis of shear stress versus shear rate of emulsions prepared with ERCA (5, 10, and 15%) was well fitted to the power law model with a high R^2^ (0.97–0.98) as shown in Table 1. Flow behavior indices (*n*) of all emulsions were significantly decreased from 0.63 to 0.52 with increasing ERCA concentrations of emulsions from 5 to 15%, respectively. Consistency index (K) and apparent viscosity (*η*_a,100_) values of emulsions were significantly increased with increasing ERCA concentration, consistent with results of Lu et al. (2018) [21] who reported that increasing the concentration of maize starch in emulsions could lead to an increase in apparent viscosity value since higher quantification of the starch in emulsions could lead to stabilization of a larger surface area. Guo et al. (2021) [22] have reported that the decrease in n value and the increase in K value with increasing concentrations of flaxseed gum in emulsions are related to the formation of stronger intermolecular networks in higher concentrations of gum. Therefore, decreased n values and increased K and apparent viscosity values with higher concentrations of ERCA found in this study could be associated with the formation of stronger intermolecular networks in the emulsions.

#### 3.1.2. Frequency Sweep

Frequency sweep results of emulsion samples are shown in Figure 2. All emulsions showed elastic behavior rather than viscous behavior because they had higher storage modulus (G′) values than loss modulus (G″) values during the increase in angular frequency. Complex viscosity (*η**) was decreased with an increase in angular frequency, indicating a typical shear thinning behavior of emulsions. These results might be attributed to network disruption of emulsions according to frequency sweep [14].

An increase in ERCA concentrations of emulsions resulted in increased values of G′, G″, and *η**. Luo et al. [14] (2019) reported that G′, G″, and *η** values of emulsions with different concentrations of citrus pectin increased with increasing concentrations of citrus pectin because citrus pectin can act as a stabilizing agent and form a gel network in emulsion. Additionally, Meng et al. (2018) [2] noted that elevating concentrations of hydroxypropyl methyl cellulose (HPMC) in emulsion can significantly increase G′ and G″ values. Increases of G′, G″, and *η** values also suggested the effect of hydrophobic group in ERCA. According to Lee & Chang (2019) [11], emulsion formed by esterified maca starch showed high emulsion stability due to new hydrophobic carbonyl groups in esterified maca starch. Esterified corn starch with octenyl succinic anhydride is an amphiphilic starch. The hydrophobic group such as carbonyl group can adsorb into oil phase while the hydrophilic group can adsorb into water phase and form a thick surface of oil droplets (Yu et al., 2019) [23].

Based on rheological results in this study, emulsions prepared with ERCA showed higher K, apparent viscosity, G′, G″, and *η** values with increasing concentration of ERCA, indicating that emulsions with higher ERCA concentrations might be more stable and that they could prevent disruption of oil droplets in emulsions.

### 3.2. Characterization of Oleogels

#### 3.2.1. FT-IR Analysis

FT-IR spectra of soybean oil, beeswax, ERCA, and oleogels are shown in Figure 3. The FT-IR spectrum of soybean oil showed the presence of characteristic peaks at 3008, 2922, 2853, 1743, 1156, 1064, and 721 cm^−1^ which indicated C–H stretching of alkene groups, CH_3_ stretching, CH_2_ stretching, C=O stretching, C–O–H stretching, C–O–C stretching, and alkyl chain, respectively [2,24].

Spectra of ERCA presented characteristic absorption bands at 3413 cm^−1^ (O–H stretching), 2929 cm^−1^ (C–H stretching), 1640 cm^−1^ (O–C–O stretching), 925 cm^−1^ (α-1,6-D-glucosidic bonds), 856 cm^−1^ (α-configuration), and 758 cm^−1^ (α-1,4-D-glucosidic bonds), consisted with previous studies [12,25]. In the spectra of ERCA, a peak at around 1734 cm^−1^ was typically associated with new ester linkages between hydroxyl groups of rice flour and carboxyl groups of citric acid [10].

For the FT-IR spectrum of beeswax, characteristic peaks were observed at 3008, 2922, 2853, 1743, 1459, and 1156 cm^−1^ which were related to the C–H stretching of the alkene groups, CH_3_ stretching, CH_2_ stretching, C=O stretching, C–O–C stretching, and C–O–H stretching, respectively [2,26].

For the FT-IR spectrum of oleogels prepared with ERCA 5%, 10%, and 15%, a new peak at 1591 cm^−1^, which are not present in the spectrum of oleogel with ERCA 0%, corresponded to C=O stretching of carboxyl groups [9]. Absorption peaks at about 1734 cm^−1^ were assigned to ester linkages of ERCA and ester linkages of soybean oil, as already mentioned previously. These results from oleogels with ERCA showed that hydroxyl groups in the rice flour were replaced with ester carbonyl and carboxyl groups from citric acid [9]. Since the hydroxyl group was replaced by the hydrophobic ester carbonyl group in starch of ERCA, the hydrophobicity of ERCA was improved [27].

Broad peaks at around 3300–3400 cm^−1^, which were not shown in the spectra of soybean oil and oleogel prepared with 0% ERCA, were observed in the spectra of oleogel samples prepared with ERCA. These broad peaks corresponded to -OH stretching vibrations attributed to intramolecular or intermolecular hydrogen bonding between ERCAs. Meng et al. (2018) [2] have reported that FT-IR spectra of oleogels with polysaccharide such as fenugreek gum, guar gum, and Arabic gum showed broad peaks at around 3300–3400 cm^−1^, which indicated the semi-crystalline structure produced by the new hydrogen bonding in oleogels. Therefore, oleogels with interface structure was formed due to hydrogen bonding between hydroxyl groups of ERCAs.

Another new peak appeared at 1018 cm^−1^ in oleogels produced with ERCA, which was related to amorphous regions of gelatinized starch [28]. It has been reported that the amorphous structure in starch formed by gelatinization can increase the thickness and density of interface layer, subsequently stabilizing the emulsion gel networks [29]. Accordingly, in this study, the hydrophobic groups of ERCA in oleogels appeared, and new peaks (hydrogen bond and amorphous regions) were shown in the spectra of oleogel prepared with ERCA (5, 10, and 15%). These new peaks did not appear in the oleogel prepared with 0% ERCA.

#### 3.2.2. Oil Loss

Oil loss is an indicator that estimates the oil binding capacity in oleogels. It is a valuable index to evaluate the quality of oleogels [24]. Effects of ERCA concentrations on oil loss of oleogel samples are shown in Table 2. It was obvious that, with increasing ERCA concentration, oil loss values of oleogels were significantly decreased. This finding could be due to the elastic interface layer formed by ERCA and the protection of oil droplets [14]. According to Naeil et al. (2020) [30], oil loss values of oleogels decreased with increasing concentrations of HPMC because stronger gel network in oleogels was formed between hydroxyl group of HPMC to increase the trapping of liquid oil phase. Results of the present study suggest that the reduction in oil leakage for oleogels produced with ERCA could be attributed to the increase in intermolecular network between ERCAs, consequently improving the trapping of liquid oil phase in oleogels. The entrapped oil in oleogels prepared with ERCA could be related to gelatinized starch in ERCA. Sjöö et al. (2015) [31] have reported that gelatinized starch in emulsion gel has an ability to create a denser layer to protect oil drop. In this study, lower leakages of oil in oleogel with higher ERCA concentrations indicated a denser and more stable interface layer of oleogels associated with the intermolecular network of ERCAs and higher hydrophobic properties of ERCA to prevent oil release.

### 3.3. Rheological Properties of Oleogels

#### 3.3.1. Time-Dependent Properties

In this study, to investigate thixotropic and structure-recovery properties of oleogels, time-dependent fluid measurements at shear rate of 0.1 → 10 → 0.1 s^−1^ were conducted. It has been reported that thixotropy is one of the most crucial rheological properties to elucidate the possible application of oleogel samples in food industry [32].

As shown in Figure 4, an increase in ERCA concentrations led to an increase in the apparent viscosity value of oleogel at a shear rate of 0.1 → 10 → 0.1 s^−1^. Apparent viscosity values of all oleogels deceased when the shear rate was raised from 0.1 to 10 s^−1^. At the highest shear rate (10 s^−1^), apparent viscosity values of oleogels were constant. When the shear rate was slowed down from 10 s^−1^ to 0.1 s^−1^, apparent viscosity values of oleogels increased, suggesting their structure-recovery properties. According to Jiang et al. (2018) [33], viscosity recovery values of oleogels above 80% indicate a good structural recovery. The structure recovery can be resulted from the occurrence of structuring units (crystallinity and polymer) which could account for the establishment of recoverable microstructure on the elimination of mechanical shear forces [32]. In this study, viscosity recovery values of oleogels prepared with ERCA at 0, 5, 10, and 15% were calculated to be about 83, 85, 85, and 100%, respectively. Furthermore, apparent viscosity values and structure recovery properties were improved with increasing concentration of ERCA due to the formation of reversible and strong gel network. Meng et al. (2018) [2] have investigated thixotropic properties of oleogels with HPMC and reported that higher concentrations of HPMC in oleogels can improve structure recovery. Results of the present study suggest that oleogels prepared with ERCA have possible applications in food that demands shear force during production because of their capabilities to recover the functionality and mechanical properties.

#### 3.3.2. Frequency Sweep

Frequency sweep results of oleogel samples are shown in Figure 5. All oleogels showed a solid-like behavior because G′ values were always higher than G″ values at all frequency regions. All oleogels also exhibited a strong gel characteristic [2]. The G′, G″, and *η** values of oleogels increased with an increase in ERCA concentration. Meng et al. (2018) [2] have reported that higher HPMC concentrations in oleogels can lead to higher G’ and G″ with increasing frequency, suggesting the formation of intra- and inter-molecular hydrogen bonding from HPMC. This phenomenon was because the interface gel network can provide mechanical barrier and increased emulsifying stability [14]. Higher increases of G′, G″, and *η** values of oleogel with increasing ERCA concentrations were also related to the gelatinization of starch in ERCA. Yulianingsih & Gohtani (2018) [34] have noted that higher concentrations of starch provide denser interfacial layer and an increase in network during gelatinization, subsequently leading to higher viscosity in starch based high internal phase gel. In this study, oleogels’ G′, G″, and *η** values were greater with increasing ERCA concentrations. Higher ERCA concentrations might have contributed to a stronger interfacial layer in oleogels.

#### 3.3.3. Temperature Sweep

Figure 6 shows temperature sweep results of oleogel prepared with ERCA during heating from 4 °C to 96 °C and subsequent cooling from 96 °C to 4 °C. Higher ERCA concentrations resulted in higher G′ and G″ values. During heating, G′ and G″ values of oleogels prepared with 0% ERCA decreased. The G′ and G″ values of oleogel prepared with 5%, 10%, and 15% ERCA decreased when the temperature was below 70 °C. However, when the temperature was above about 70 °C, G′ and G″ values increased because of swelling and gelatinization of starch in ERCA.

The G′ and G″ values of oleogels prepared with 0% ERCA increased with a reduction in temperature from 96 °C to 4 °C. Meanwhile, G′ and G″ values of oleogel prepared with ERCA at 5%, 10%, and 15% could also be increased due to the retrogradation of starch in ERCA [35] and recrystallization of beeswax in oleogel [36]. In this study, G′ and G″ values of oleogels were elevated with increasing ERCA concentrations under heating and cooling.

Based on results of oil loss, time-dependent fluid test, frequency sweep test, and temperature sweep test, two mechanisms could account for the decrease in oil loss values and the enhancement in rheological characteristics such as apparent viscosity, G′, G″, and *η** values of oleogels prepared with ERCA.

First, hydrophobic ester carbonyl groups of ERCA could lead to a reduction in oil loss and an improvement of rheological properties. According to Zhou et al. (2016) [37], waxy maize starch modified with citric acid can be used as an emulsion stabilizer because hydroxyl groups in the modified starch are substituted by hydrophobic carbonyl groups, subsequently leading to improved emulsion stability. Yu et al. (2019) [23] have reported that hydrophobic groups can lead to the formation of a thick layer at the surface of oil droplets. Fonseca-Florido et al. (2018) [9] have reported that the emulsion gel formed by esterified corn starch with octenyl succinic acid shows higher apparent viscosity and higher G′ and G″ values compared with the emulsion gel formed by native corn starch because hydrophobic carbonyl groups of esterified corn starch can enhance the stability of emulsion. In the present study, hydrophobic ester carbonyl groups in oleogels prepared with ERCA were confirmed as shown in FT-IR spectra (Figure 3). Therefore, hydrophobic carbonyl groups of ERCA in oleogels enhanced emulsion stability, subsequently, causing not only the reduction in oil leaking, but also an improvement of rheological properties of oleogels.

Secondly, decreased values of oil loss and increased values of rheological properties for oleogels with ERCA can be explained by the introduction of new intermolecular and intramolecular hydrogen bonds between hydroxyl groups of ERCA during swelling and gelatinization. In the present study, as already shown in FT-IR spectra (Figure 3), new characteristic peaks at around 3300–3400 cm^−1^ and new peaks at around 1018 cm^−1^ were found in olegels prepared with ERCA (5, 10, and 15%) which indicated intermolecular and intramolecular hydrogen bonds between hydroxyl groups of ERCA and amorphous regions in oleogels prepared with ERCA, respectively. Gelatinization is a transformation of the crystalline structure of starch granules to an amorphous state in the presence of heat and water [38]. According to Sjöö et al. (2015) [31], during heat treatment with water, starch granules can swell and/or gelatinize and finally form a cohesive barrier with thicker and denser layer at the oil-water interface. Meng et al. (2018) [2] have found new hydrogen bonds in oleogels prepared with HPMC and reported that intramolecular and/or intermolecular hydrogen bonding in oleogels observed in FT-IR spectra formed a relatively orderly structure with increased values of apparent viscosity, G′, and G″. In summary, the reduction in oil loss and the increase in rheological properties of oleogels with ERCA can be associated with an increased emulsion stability due to hydrophobic carbonyl groups and thicker and denser layer formed by hydrogen bonds in oleogels.

### 3.4. Oxidative Stability of Oleogels during Heat Treatment

Oxidative stability of oleogel was evaluated during heat treatment at 140 °C for 2 h. Öğütcü et al. (2015) [39] reported that heat treatment at 140 °C is a very critical factor in enhancing lipid oxidation. Peroxide value (PV) reflects primary oxidation products and the oxidative state of unsaturated fats in the early phase. The *p*-Anisidine value (AV) is an indicator of secondary oxidation products of primary oxidation products. TOTOX value indicates high temperature oxidative performance [40].

Figure 7 shows changes in peroxide value, *p*-anisidine value, and TOTOX value of soybean oil and oleogels prepared with different concentrations of ERCA during heating at 140 °C for 0, 1, and 2 h. After all samples were heated at 140 °C for 2 h, PV, AV, and TOTOX values of oleogels were much lower than those of soybean oil. In particular, the oleogel prepared with 15% ERCA showed the lowest PV, AV, and TOTOX values. Luo et al. (2019) [14] have noted that the oleogelation of camellia oil with internal structuring (tea polyphenol-palmitate) and external coating (citrus pectin) resulted in the prevention of oxidation of the oil because the oil was entrapped in the oleogel structure formed by external coating with citrus pectin, subsequently, inhibiting the exposure of the oil and slowing down oxidation by air. Lee et al. (2019) [4] have also reported that the oxidative stability of fish oil in oleogels internally structured by beeswax and externally coated by whey protein isolate is improved compared to unencapsulated fish oil during heat treatment. In addition, the physical barrier layer of swollen starch can prevent oxidation during heat treatment [31]. Therefore, lower PV, AV, and TOTOX values of oleogels prepared with ERCA might be associated with the external coating and the three-dimensional structure formed by ERCA which can act as a physical barrier for air, subsequently enhancing oxidative stability.

### 3.5. Application of Oleogels to Cookies

#### Physicochemical Properties of Cookies with Oleogels Prepared with ERCA

In this study, oleogels prepared with ERCA were used to investigate the possibility of using them to replace high saturated fat products such as commercial shortenings in food processing. Figure 8 shows the appearance of a control cookie with commercial shortening and cookies with oleogels prepared with different concentrations of ERCA. The overall appearance of all cookies was not remarkably different. Furthermore, the spread factor and loss rate of control cookie were not significantly different from those of cookies prepared with oleogels (Table 3). Zhao et al. (2020) [6] have also noted that the addition of corn oil based oleogel to cookie samples for replacing shortening does not considerably affect the weight, width, thickness, or spread factor of the cookies.

Fatty acids compositions of control cookie and cookie prepared using oleogel with 15% ERCA are presented in Table 4. The highest component fatty acid in the control cookie was palmitic acid (C16:0) while the cookie prepared with oleogel was rich in oleic acids (C18:1). The palmitic acid was found to be the main fatty acid component of shortening [41].

Proportions of total saturated and unsaturated fatty acids in control cookie were found to be 52.92% and 46.77%, respectively. The cookie with oleogel prepared with 15% ERCA showed lower contents of saturated fatty acids but higher contents of unsaturated fatty acids than the control cookie. Therefore, replacing shortening by oleogels prepared with ERCA did not considerably affect quality properties such as appearance and spread factor. However, it improved the content of unsaturated fatty acids in cookies.

## 4. Conclusions

In this study, oleogels for replacing fats were prepared using different concentrations of esterified rice flour with citric acid (ERCA). ERCA was used as an emulsifier and coating material for emulsions and oleogels. The apparent viscosity (*η*_a,100_), consistency index (K), storage modulus (G′), loss modulus (G″), and complex viscosity (*η**) of emulsions were increased with increasing concentration of ERCA. Results obtained from FT-IR spectra, oil loss test, and rheological properties (time-dependent fluid, frequency sweep, and temperature sweep tests) suggest that the addition of ERCA to oleogels can reduce oil loss and improve rheological properties (such as apparent viscosity, G′, G″, and *η** values) due to an increased emulsion stability resulting from hydrophobic carbonyl groups in ERCA of oleogel and the thicker and denser layer formed by the introduction of new intermolecular and intramolecular hydrogen bonds between hydroxyl groups of ERCA. Finally, it was found that ERCA could be used as a fat replacer to produce cookies without causing a considerable decrease in quality properties.

## Figures and Tables

**Figure 1 foods-11-00242-f001:**
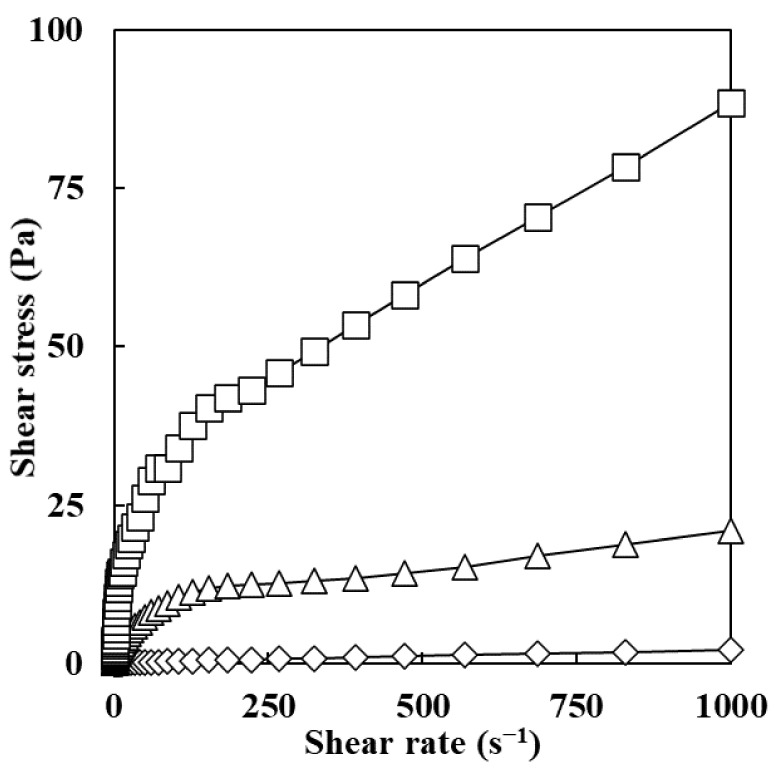
Shear stress vs. shear rate for emulsion prepared with different concentrations of esterified rice flour with citric acid (ERCA). (◇) emulsion prepared with 5% ERCA, (Δ) emulsion prepared with 10% ERCA, and (□) emulsion prepared with 15% ERCA.

**Figure 2 foods-11-00242-f002:**
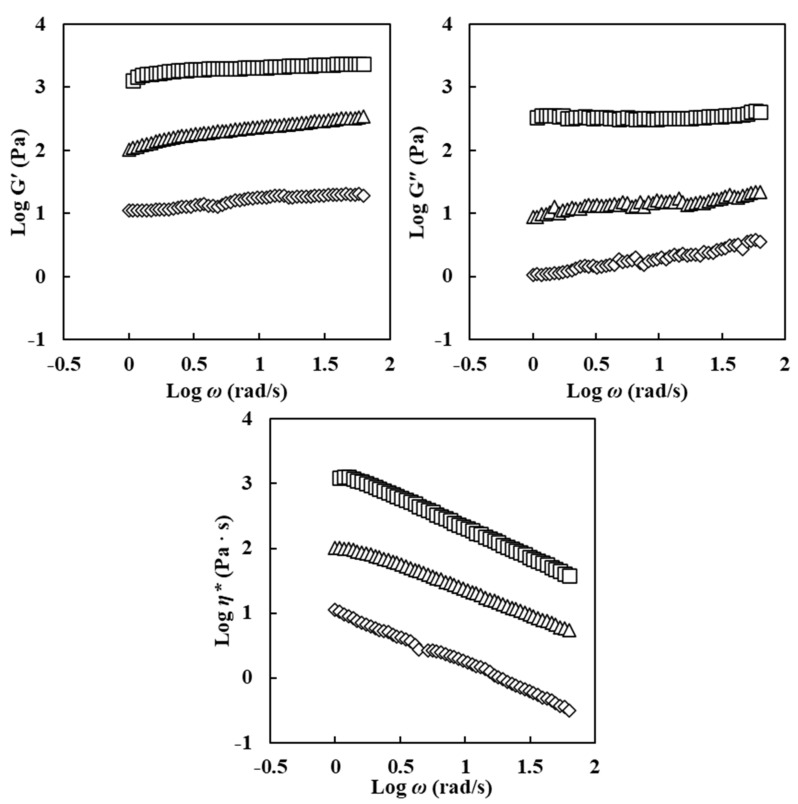
Log storage modulus (G′), log loss modulus (G″), and log complex viscosity (*η**) vs. log angular frequency (*ω*) for emulsion prepared with different concentrations of esterified rice flour with citric acid (ERCA). (◇) emulsion prepared with 5% ERCA, (Δ) emulsion prepared with 10% ERCA, and (□) emulsion prepared with 15% ERCA.

**Figure 3 foods-11-00242-f003:**
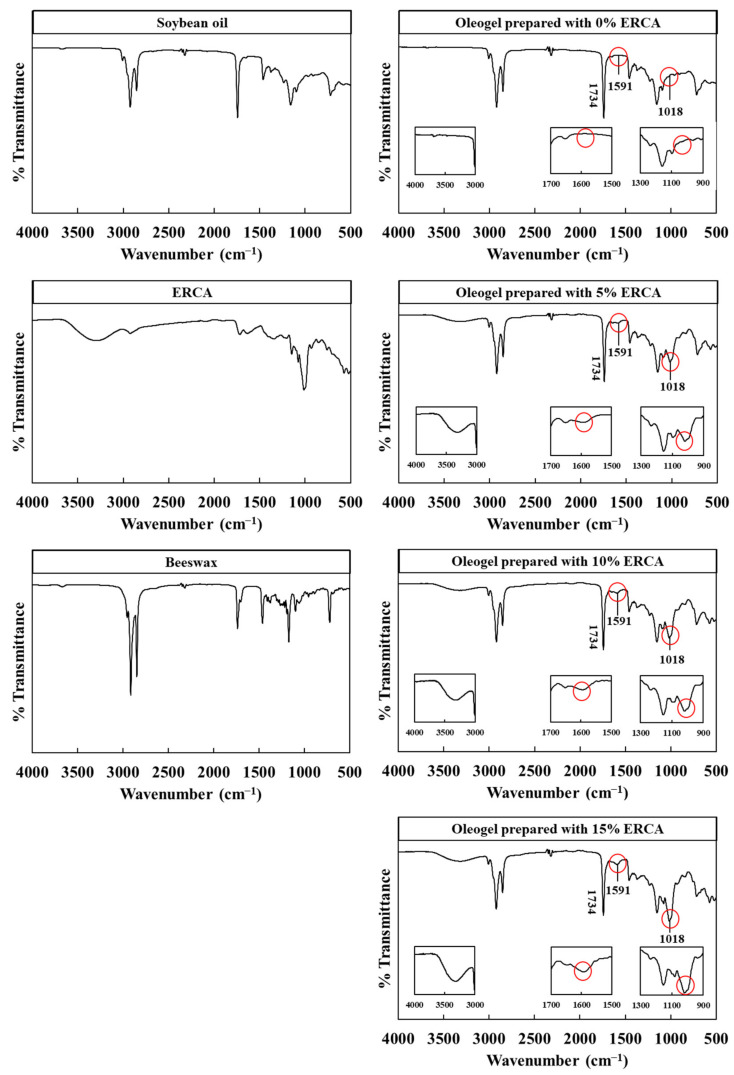
FT-IR spectra of soybean oil, beeswax, esterified rice flour with citric acid (ERCA), and oleogels prepared with different concentrations of ERCA (0, 5, 10, and 15%, respectively).

**Figure 4 foods-11-00242-f004:**
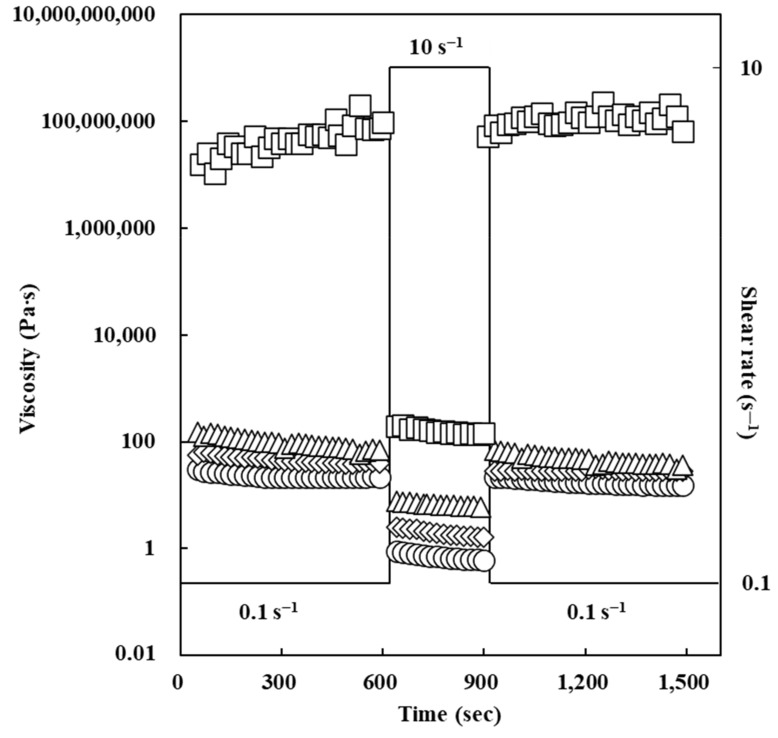
Time-dependent behavior of oleogels prepared with different concentrations of esterified rice flour with citric acid (ERCA). (○) Oleogel prepared with 0% ERCA, (◇) Oleogel prepared with 5% ERCA, (Δ) Oleogel prepared with 10% ERCA, and (□) Oleogel prepared with 15% ERCA.

**Figure 5 foods-11-00242-f005:**
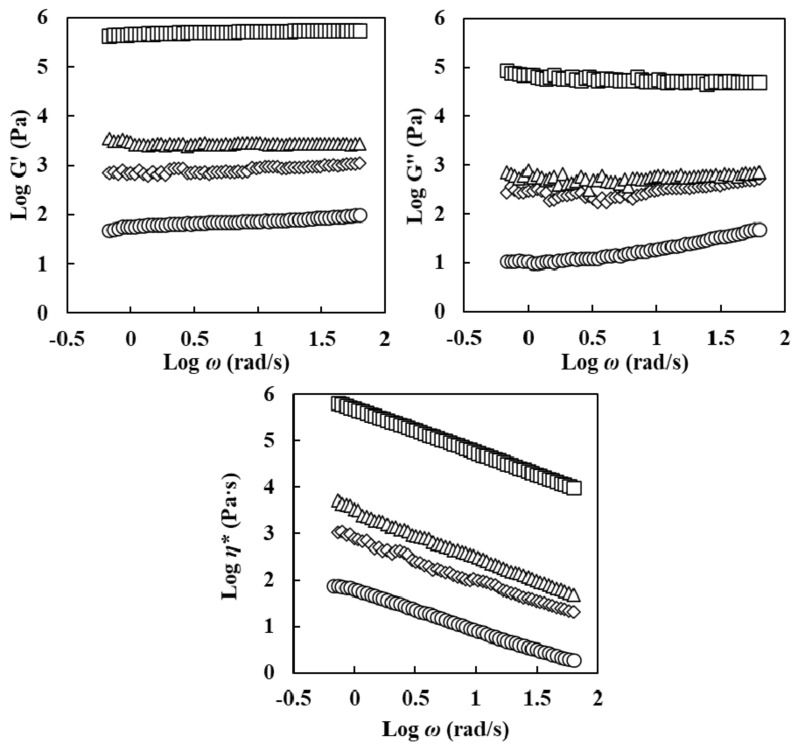
Log storage modulus (G′), log loss modulus (G″), and log complex viscosity (*η**) vs. log angular frequency (*ω*) for oleogels prepared with different concentrations of esterified rice flour with citric acid (ERCA). (○) Oleogel prepared with 0% ERCA, (◇) Oleogel prepared with 5% ERCA, (△) Oleogel prepared with 10% ERCA, and (□) Oleogel prepared with 15% ERCA.

**Figure 6 foods-11-00242-f006:**
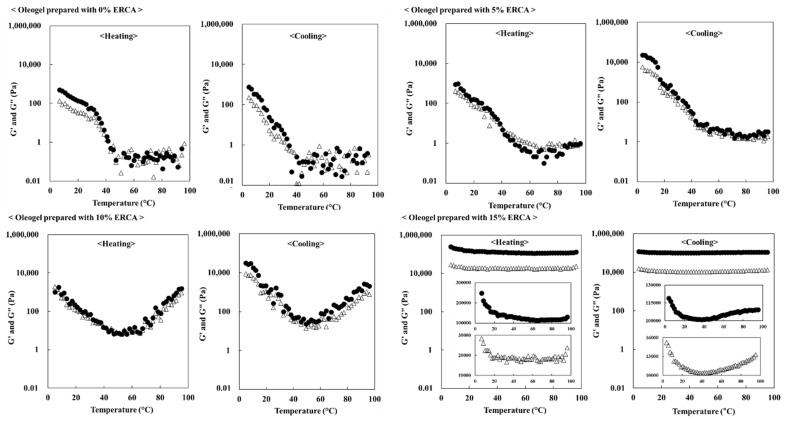
Temperature dependence of storage modulus (G′: ●) and loss modulus (G″: Δ) for oleogels prepared with different concentrations of esterified rice flour with citric acid (ERCA; 0, 5, 10, and 15%, respectively) during heating from 4 to 96 °C and subsequent cooling from 96 to 4 °C at a rate of 5 °C/min.

**Figure 7 foods-11-00242-f007:**
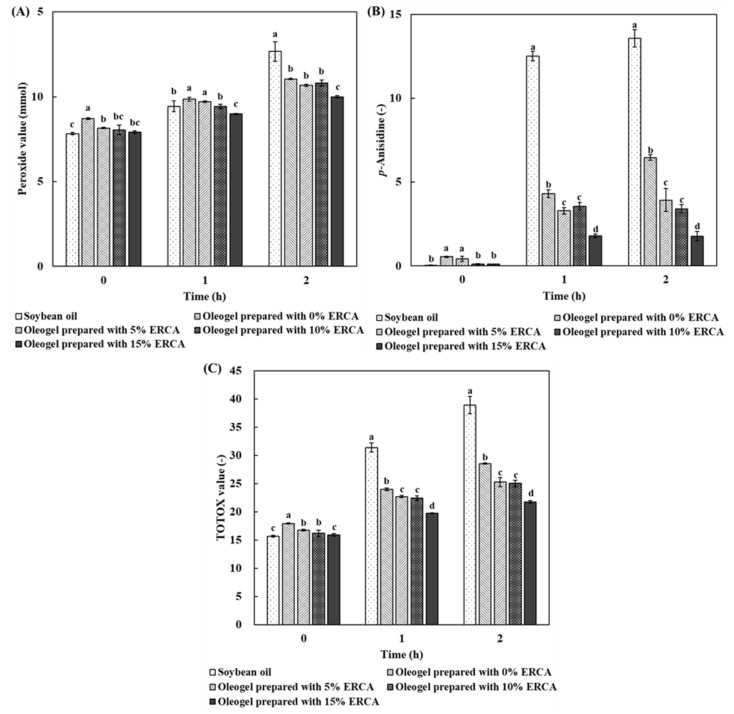
Changes in peroxide value (**A**), *p*-Anisidine value (**B**), and TOTOX value (**C**) of soybean oil and oleogels prepared with different concentrations of esterified rice flour with citric acid (ERCA; 0, 5, 10, and 15%, respectively) during heating at 140 °C for 0, 1, and 2 h. Bars with different letters within the same bar differ significantly (*p* < 0.05).

**Figure 8 foods-11-00242-f008:**
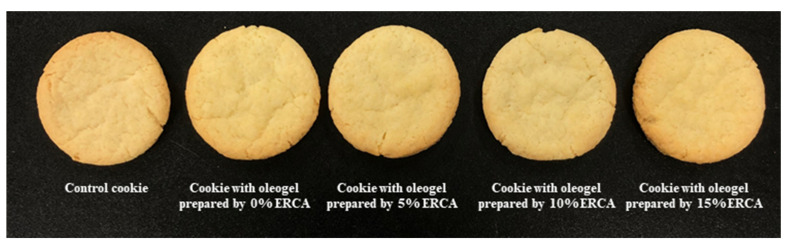
Appearance of control cookie and cookie with oleogel prepared by different concentrations of esterified rice flour with citric acid (ERCA; 0, 5, 10, and 15%, respectively).

**Table 1 foods-11-00242-t001:** Steady shear rheological properties of emulsion prepared with different concentrations of esterified rice flour with citric acid (ERCA).

Samples	Flow Behavior Index *n* (−)	Consistency Index K (Pa·s^n^)	Apparent Viscosity *η*_a, 100_ (Pa·s)	R^2^
Emulsion prepared with 5% ERCA	0.63 ± 0.02 ^a1^	0.18 ± 0.01 ^c^	0.03 ± 0.00 ^c^	0.98
Emulsion prepared with 10% ERCA	0.57 ± 0.01 ^b^	0.82 ± 0.06 ^b^	0.11 ± 0.00 ^b^	0.97
Emulsion prepared with 15% ERCA	0.52 ± 0.02 ^c^	1.71 ± 0.05 ^a^	0.19 ± 0.02 ^a^	0.98

^1^ Values with different letters within the same column differ significantly (*p* < 0.05).

**Table 2 foods-11-00242-t002:** Oil loss of oleogels prepared with different concentrations of esterified rice flour with citric acid (ERCA).

Samples	Oil Loss (%)
Oleogel prepared with 0% ERCA	49.83 ± 0.89 ^a1^
Oleogel prepared with 5% ERCA	15.53 ± 2.88 ^b^
Oleogel prepared with 10% ERCA	2.19 ± 0.89 ^c^
Oleogel prepared with 15% ERCA	0.33 ± 0.32 ^d^

^1^ Values with different letters within the same column differ significantly (*p* < 0.05).

**Table 3 foods-11-00242-t003:** Spread factor and loss rate of control cookie and cookie with oleogel prepared by different concentrations of esterified rice flour with citric acid (ERCA; 0, 5, 10, and 15%, respectively).

Samples	Spread Factor	Loss Rate
Control cookie	4.64 ± 0.17 ^a1^	17.58 ± 0.29 ^a^
Cookie with oleogel prepared by 0% ERCA	4.65 ± 0.24 ^a^	17.09 ± 1.24 ^a^
Cookie with oleogel prepared by 5% ERCA	4.51 ± 0.25 ^a^	16.63 ± 0.62 ^a^
Cookie with oleogel prepared by 10% ERCA	4.55 ± 0.31 ^a^	16.44 ± 0.09 ^a^
Cookie with oleogel prepared by 15% ERCA	4.37 ± 0.23 ^a^	17.72 ± 1.52 ^a^

^1^ Values with different letters within the same column differ significantly (*p* < 0.05).

**Table 4 foods-11-00242-t004:** Compositions of fatty acids in control cookie and cookie prepared with oleogel prepared by 15% esterified rice flour with citric acid (ERCA).

Fatty Acids (%)	Control Cookie	Cookie with Oleogel Prepared by 15% ERCA
C4:0	0.00 ± 0.00 ^a1^	0.00 ± 0.00 ^a^
C6:0	0.04 ± 0.00 ^a^	0.03 ± 0.00 ^a^
C8:0	0.40 ± 0.01 ^a^	0.24 ± 0.01 ^b^
C10:0	0.31 ± 0.00 ^a^	0.19 ± 0.00 ^b^
C11:0	0.00 ± 0.00 ^a^	0.00 ± 0.00 ^a^
C12:0	2.34 ± 0.02 ^a^	1.43 ± 0.01 ^b^
C13:0	0.00 ± 0.00 ^b^	0.05 ± 0.01 ^a^
C14:0	1.86 ± 0.01 ^a^	1.18 ± 0.01 ^b^
C14:1	0.03 ± 0.00 ^a^	0.02 ± 0.00 ^a^
C15:0	0.07 ± 0.00 ^a^	0.05 ± 0.00 ^b^
C15:1	0.00 ± 0.00 ^a^	0.00 ± 0.00 ^a^
C16:0	41.91 ± 0.22 ^a^	30.24 ± 0.37 ^b^
C16:1	0.28 ± 0.01 ^a^	0.20 ± 0.00 ^b^
C17:0	0.18 ± 0.01 ^a^	0.14 ± 0.01 ^b^
C17:1	0.00 ± 0.00 ^a^	0.00 ± 0.00 ^a^
C18:0	5.22 ± 0.01 ^a^	4.93 ± 0.04 ^b^
C18:1	35.44 ± 0.13 ^a^	31.11 ± 0.12 ^b^
C18:2	10.54 ± 0.05 ^b^	25.96 ± 0.17 ^a^
C18:3	0.28 ± 0.01 ^b^	2.41 ± 0.02 ^a^
C20:0	0.37 ± 0.00 ^a^	0.38 ± 0.01 ^a^
C20:1	0.18 ± 0.01 ^b^	0.38 ± 0.01 ^a^
C20:2	0.03 ± 0.01 ^a^	0.04 ± 0.00 ^a^
C21:0	0.01 ± 0.01 ^b^	0.03 ± 0.01 ^a^
C22:0	0.10 ± 0.00 ^b^	0.25 ± 0.01 ^a^
C23:0	0.02 ± 0.01 ^b^	0.04 ± 0.00 ^a^
C24:0	0.10 ± 0.00 ^b^	0.14 ± 0.00 ^a^
Saturated	52.92 ± 0.26 ^a^	39.33 ± 0.45 ^b^
Unsaturated	46.77 ± 0.18 ^b^	60.12 ± 0.32 ^a^

^1^ Values with different letters within the same row differ significantly (*p* < 0.05).

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
