# Peer review of "Rheological and Physicochemical Properties of Oleogel with Esterified Rice Flour and Its Suitability as a Fat Replacer"

_foods, 2022, doi:10.3390/foods11020242_

Round 1

Reviewer 1 Report

  1. The oleogel was prepared using emulsion, but the emulsion information is not mentioned in the abstract. Therefore, the authors should rewrite the abstract.
  2. Why did the author choose 140 ËšC instead of 60 ËšC for thermal treatment in accelerated oxidation? Please clarify the basis by which 140 ËšC was selected and tested.

Author Response

  1. The oleogel was prepared using emulsion, but the emulsion information is not mentioned in the abstract. Therefore, the authors should rewrite the abstract.

  -> Based on your comment, we inserted the information on the emulsions and revised the abstract.

  1. Why did the author choose 140 ËšC instead of 60 ËšC for thermal treatment in accelerated oxidation? Please clarify the basis by which 140 ËšC was selected and tested.

-> According to the previous literatures, heat treatment at 140°C is a very critical factor in enhancing lipid oxidation.  Therefore, we chose 140 ËšC instead of 60 ËšC.  Please see Lines 442 and inserted the following references.

 1) ÖÄŸütcü, Mustafa, Nazan ArifoÄŸlu, and Emin Yılmaz. "Storage stability of cod liver oil organogels formed with beeswax and carnauba wax." International Journal of Food Science & Technology 50.2 (2015): 404-412.

 2) Gravelle, Andrew J., Shai Barbut, and Alejandro G. Marangoni. "Ethylcellulose oleogels: Manufacturing considerations and effects of oil oxidation." Food Research International 48.2 (2012): 578-583.

Reviewer 2 Report

The manuscript titled “Rheological and physicochemical properties of oleogel with esterified rice flour and its suitability as fat replacer” deals within the scope of the Foods Journal, by investigating an interesting topic of research. The quality of the presented research work is very good and represents valuable research results of the application of olegels as fat replacer in biscuit production.

Corrections to be made

Line 57: “…increase the rheological properties…”. Maybe it's better to use the phrase “…improve the rheological properties…”.

Line 73: This sentence is a little confusing: “…citric acid anhydrous (30% of rice flour, dry basis) was mixed with distilled water...” How much citric acid is 30% of rice flour? Is it 30% of 50 g? How much water was used? Please rephrase this sentence to be more understandable.

Line 169: What were the ratio of oleogel and shortening in the blend?

Line 467: It is better to use “control cookie” instead of “normal cookie”. Please, correct this in the rest of the text accordingly.

Author Response

  1. Line 57: “…increase the rheological properties…”. Maybe it's better to use the phrase “…improve the rheological properties…”.

-> We revised from “increase” to “improve”.  Please see line 61.

  1. Line 73: This sentence is a little confusing: “…citric acid anhydrous (30% of rice flour, dry basis) was mixed with distilled water...” How much citric acid is 30% of rice flour? Is it 30% of 50 g? How much water was used? Please rephrase this sentence to be more understandable.

-> We added the amount of rice flour and water.  Please see lines 77.

  1. Line 169: What were the ratio of oleogel and shortening in the blend?

-> We used the ratio at the ratio of 50:50.  Please see lines 174.

  1. Line 467: It is better to use “control cookie” instead of “normal cookie”. Please, correct this in the rest of the text accordingly.

-> We changed from “normal cookie” to “control cookie” in the entire manuscript and Figure 8.